# Acquired Evolution of Mitochondrial Metabolism Regulated by HNF1B in Ovarian Clear Cell Carcinoma

**DOI:** 10.3390/cancers13102413

**Published:** 2021-05-17

**Authors:** Ken Yamaguchi, Sachiko Kitamura, Yoko Furutake, Ryusuke Murakami, Koji Yamanoi, Mana Taki, Masayo Ukita, Junzo Hamanishi, Masaki Mandai

**Affiliations:** 1Department of Gynecology and Obstetrics, Graduate School of Medicine, Kyoto University, Kyoto 606-8507, Japan; schan109@kuhp.kyoto-u.ac.jp (S.K.); yfurutake@kuhp.kyoto-u.ac.jp (Y.F.); ryusukem@kuhp.kyoto-u.ac.jp (R.M.); kojiymni@kuhp.kyoto-u.ac.jp (K.Y.); takimana@kuhp.kyoto-u.ac.jp (M.T.); ukita@kuhp.kyoto-u.ac.jp (M.U.); jnkhmns@kuhp.kyoto-u.ac.jp (J.H.); mandai@kuhp.kyoto-u.ac.jp (M.M.); 2Department of Gynecology, Shiga General Hospital, Moriyama, Shiga 524-8524, Japan

**Keywords:** ovarian clear cell carcinoma, metabolism, mitochondrion, Warburg effect, glutathione synthesis

## Abstract

**Simple Summary:**

Ovarian clear cell carcinoma (CCC) exhibits unique characteristics, including slow growth, glycogen accumulation in the cytoplasm, and poor prognosis for stress resistance. Several molecular targeting agents have failed to treat ovarian CCC. Recent reports have identified metabolic alterations through HNF1B, which is highly expressed in ovarian CCC. The Warburg effect, GSH synthesis, and mitochondrial regulation occur in CCC. The metabolic behaviors of ovarian CCC resemble the evolution of life to survive in stressful environments. Understanding the fundamental biology of ovarian CCC might help in the development of novel therapeutic strategies.

**Abstract:**

Clear cell carcinoma (CCC) of the ovary exhibits a unique morphology and clinically malignant behavior. The eosinophilic cytoplasm includes abundant glycogen. Although the growth is slow, the prognosis is poor owing to resistance to conventional chemotherapies. CCC often arises in endometriotic cysts and is accompanied by endometriosis. Based on these characteristics, three clinical questions are considered: why does ovarian cancer, especially CCC and endometrioid carcinoma, frequently occur in endometriotic cysts, why do distinct histological subtypes (CCC and endometrioid carcinoma) arise in the endometriotic cyst, and why does ovarian CCC possess unique characteristics? Mutations in *AT-rich interacting domain-containing protein 1A* and *phosphatidylinositol-4,5-Bisphosphate 3-Kinase Catalytic Subunit alpha* genes may contribute to the carcinogenesis of ovarian CCC, whereas *hepatocyte nuclear factor-1-beta* (*HNF1B*) plays crucial roles in sculpting the unique characteristics of ovarian CCC through metabolic alterations. HNF1B increases glutathione synthesis, activates anaerobic glycolysis called the Warburg effect, and suppresses mitochondria. These metabolic changes may be induced in stressful environments. Life has evolved to utilize and control energy; eukaryotes require mitochondria to transform oxygen reduction into useful energy. Because mitochondrial function is suppressed in ovarian CCC, these cancer cells probably acquired further metabolic evolution during the carcinogenic process in order to survive stressful environments.

## 1. Introduction

Ovarian cancer is the worst gynecologic malignancy and includes heterogeneous histological and biomolecular features [1,2,3]. Clear cell carcinoma (CCC) of the ovary is the second most common ovarian cancer, after high-grade serous carcinoma in Japan, and exhibits unique characteristics, including endometriosis, slow growth, association with thromboembolism, glycogen accumulation in the cytoplasm, and poor prognosis in advanced cases due to chemoresistance [4,5,6,7,8,9,10]. The unique epidemiology and clinical behavior of ovarian CCC raise three clinical questions. First, why does ovarian cancer, especially CCC and endometrioid carcinoma, occur frequently in endometriotic cysts? Second, why is endometriosis-related ovarian cancer more associated with specific histological subtypes, including CCC and endometrioid carcinoma, and why do distinct histological subtypes arise in endometriotic cysts? Third, why does ovarian CCC have unique characteristics? Recently, the molecular biological features of CCC have been reported using next-generation sequencing technology. Mutations in *AT-rich interacting domain-containing protein 1A* (*ARID1A*) and *phosphatidylinositol-4,5-Bisphosphate 3-Kinase Catalytic Subunit alpha* (*PIK3CA*) genes have been found in approximately half of the cases [11,12,13]. There are single-nucleotide polymorphisms (rs11651755) in the *hepatocyte nuclear factor-1-beta* (*HNF1B*) gene, which is highly expressed in ovarian CCC [14,15]. In addition, a detailed mutation status during each process of endometriosis-related carcinogenesis has been reported [11,12,13]. Interestingly, there are similar gene mutations in the orthotopic endometrium and ovarian endometriosis [16,17], and the possibility of carcinogenesis due to two hits in the ovary, as the same mutation has been found in eutopic endometrium, endometriotic cyst, and ovarian CCC [16]. Thus, the development of CCC has rapidly been understood from genetic analysis; however, the reasons for CCC being more frequent in endometriosis-related ovarian cancer and its unique characteristics are not clear. This review describes the roles of the carcinogenic environment leading to the development of CCC in endometriotic cysts and highlights the importance of metabolic regulation in sculpting the unique biology of ovarian CCC. Regulation of metabolism contributes to energy homeostasis from the onset of life. The tumor microenvironment can be considered a battlefield between cancer and host defense, competing for energy supply and metabolism [18]. Because mitochondria have been fundamental in evolution to produce energy, cancer cells utilize mitochondria for survival. In this review, we hypothesize that the evolution of ovarian CCC is a reason for the metabolic alterations that occur in ovarian CCC.

## 2. Endometriotic Cyst as a Carcinogenic Environment for Ovarian Cancer

Endometriosis is one of the most common gynecological disorders, affecting approximately 15% of women in their reproductive years [19]. Although endometriosis is a benign condition dependent on estrogen, endometriosis is a risk factor for ovarian cancer [20]. Although 1.3% of women in the general female population will develop ovarian cancer in their lifetime, the lifetime risk of ovarian cancer among women with endometriosis is 1.8% [21]. Large epidemiological studies showed an increased risk for ovarian cancer in patients who were hospitalized for endometriosis (SIR 1.9, 95% CI 1.3–2.8, and SIR 1.43, 95% CI 1.49–1.71, respectively) [20]. Genomic sequencing was used to identify cancer-associated mutations in endometriosis. Known somatic cancer-driver mutations in *ARID1A*, *PIK3CA*, and *KRAS* have been identified in deep endometriosis, indicating that endometriosis is a neoplastic disease [22]. Single-nucleotide polymorphisms (rs11651755) in *HNF1B* were found to be associated with endometriosis and modified the risk of ovarian cancer, as well as endometriosis [14,15]. These findings indicate that endometriosis is a precursor and a neoplastic disease.

Why does ovarian cancer develop in endometriotic cysts? Suda et al. reported that endometriosis and uterine endometrium frequently exhibit cancer-associated somatic mutations, including *KRAS* mutations and *PIK3CA* mutations, although these epithelia were histologically benign and normal [17]. Clonal expansion of epithelial cells with cancer-associated mutations is observed in endometriosis, whereas the genomic architecture of epithelial cells in the uterine endometrium is heterogeneous [17]. The burden of cancer-driver mutations, which are carried frequently in normal endometrial glands, increases with age and decreases with parity [23]. In addition to a single *ARID1A* loss-of-function mutation in an ovarian endometriotic cyst, further driver events may be needed for the malignant transformation of ovarian endometriotic cysts with *ARID1A* loss-of-function mutations [16]. These studies, using genomic assessments, indicate that part of the ovarian endometriotic cyst is a precursor of endometriosis-related ovarian cancer. Endometriosis is ectopic, in which the endometrial glands are located outside the uterine endometrium. When endometriosis occurs in the ovaries, it generates a cyst cavity by repeated bleeding, leading to the formation of an endometriotic cyst. Hemorrhage and hemosiderin deposition are often histologically identified in endometriotic cysts [24]; therefore, the tumor environment is crucial for the carcinogenesis in endometriotic cysts. An endometriotic cyst includes hemorrhage in the cyst cavity by repeated bleeding during the menstrual cycle and has quite distinct contents compared with other benign cysts. Hemorrhage and hemosiderin include iron, which causes oxidative stress through the production of reactive oxygen species (ROS) through the Fenton reaction [25,26]. Dietary iron is a risk factor for gastrointestinal cancers [27,28] and increased body iron stores have been reported to be associated with poor prognosis of several human malignant neoplasms [29,30,31,32]. High body iron stores increase the risk of cancer development through oxidative stress in several organs [33,34]. The content of endometriotic cysts contains a high concentration of free iron, and endometriotic cysts are an environment rich in oxidative stress compared to other benign cysts [24]. We confirmed the high frequency of DNA damage induced by exposure to free iron and the contents of endometriotic cysts [24]; therefore, we believe that persistent exposure to highly concentrated free iron is a possible carcinogenic environment in endometriotic cysts, possibly via the production of oxidative stress.

## 3. Two Hypothetical Theories for Carcinogenesis Process among Endometriosis-Related Ovarian Cancer

The second question we sought to address is why CCC and endometrioid carcinoma are more frequent in endometriosis-related ovarian cancer than high-grade serous carcinoma and mucinous carcinoma. The rate of co-existence of endometriosis is higher in CCC and endometrioid carcinoma than in high-grade serous carcinoma and mucinous carcinoma [20,29]. Another question is why different histological subtypes arise in endometriotic cysts. Immunohistochemistry identified high expression of HNF1B and low expression of low estrogen receptor alpha (ERa) in ovarian CCC, whereas high expression of ERa and low expression of HNF1B were observed in endometrioid carcinoma [30,31]. ERa expression was associated with a favorable prognosis in endometrioid carcinoma cases, whereas no significant association was found for CCC [31]. These findings suggest a different role for estrogen dependency between endometrioid carcinoma and CCC.

Using expression microarrays of ovarian cancer, we identified a gene set composed of 320 genes which are specific to ovarian CCC and named them ovarian CCC signatures [32]. These genes include many genes related to *HNF1B* signaling. We assessed whether the signature is involved in the stressful environment of carcinogenesis in endometriotic cysts. When cells were cultured with stressful factors, such as the contents of endometriotic cysts and iron, the ovarian CCC signature was upregulated [32], suggesting that genes specific for ovarian CCC are induced under a stressful carcinogenic environment. Recently, ferroptosis, which is a cell death caused by lipid ROS in an iron-dependent manner, was identified and is considered to be different from apoptosis and necrosis. Ferroptosis is induced by erastin, through the reduction of glutathione levels by directly inhibiting the cystine/glutamate antiporter system Xc−activity in ectopic endometrial stromal cells, whereas the influence of erastin on normal endometrial stromal cells was only slight [35]. Our previous report identified that immunohistochemistry using 8-hydroxy-2’-deoxyguanosine (8-OHdG), a marker of oxidative stress-induced DNA damage, was higher in CCC than in other histological subtypes [24], suggesting that ovarian CCC can grow in oxidative stress environments. We also evaluated the lipid ROS damage marker to evaluate the ferroptosis status. Our preliminary data showed that the lipid damage marker tended to be stronger in endometrioid carcinoma than in CCC (data not yet published). These findings indicate that CCC exhibits different responses against oxidative stress from endometrioid carcinoma, although both subtypes develop in an oxidative stress environment, suggesting that ferroptosis is suppressed in CCC. Zou et al. reported that inhibition of glutathione peroxidase 4 (GPX4) exhibits the highest selectivity and potency in killing renal and ovarian CCC cells, suggesting an intrinsic vulnerability to ferroptosis [36]. In their study, they identified that cells with clear cell morphology exhibit GPX4 dependency. These findings indicate that CCC requires distinct mechanisms to survive ferroptosis compared to other histological subtypes.

Based on the above, we hypothesized two models of pathogenesis of carcinogenesis in endometriosis-associated ovarian cancer [37]. The first mechanism is strongly influenced by the unique microenvironment of the endometriotic cysts. Persistent exposure to oxidative stress induced in iron-rich environments results in the carcinogenesis of ovarian CCC. The other mechanism is estrogen-dependent carcinogenesis, which primarily results in endometrioid carcinoma.

## 4. Unique Characteristics of Ovarian CCC Induced in Carcinogenic Environments

The third question is why CCC possesses unique characteristics, including glycogen accumulation in the cytoplasm, slow growth, and chemoresistance. The genomic alterations in endometriosis-related ovarian cancer are distinct from those induced by free iron. Free iron frequently induces *p16* gene mutations in a rat renal carcinogenesis model [38,39,40], whereas approximately half of ovarian CCC cases possess *ARID1A* and *PIK3CA* mutations [11,12,13]. These findings suggest that gene mutations in iron-rich environments are not the reason for the initiation of carcinogenesis in endometriotic cysts, although iron-induced oxidative stress is an environment that affects carcinogenesis. The ovarian CCC signature includes many genes related to oxidative stress, glycogen (glucose metabolism), coagulation, and *HNF1B* signaling [32]. Oxidative stress, glucose metabolism, and coagulation are consistent with the characteristics of ovarian CCC, indicating that the ovarian CCC signature reflects the unique biology of ovarian CCC. In addition, *HNF1B* signaling, which forms a network in the ovarian CCC signature, may be involved in ovarian CCC properties. HNF1B expression is associated with physiopathological cytoplasmic glycogen accumulation in ovarian CCC, endometrial CCC, and non-neoplastic endometrium in the mid-to-late secretory phase of the menstrual cycle and gestation [41]. A bioinformatics assessment also identified an association between HNF1B, glycogen accumulation, and thrombosis [32,42]. As shown in the previous section, the stressful carcinogenic environment in endometriotic cysts may induce the unique biology of ovarian CCC. These results suggest that HNF1B is a key factor in regulating the features of ovarian CCC; we will describe the role of HNF1B in a later section. Therefore, our next hypothesis is that the stressful carcinogenic environment in endometriotic cysts influences the biology of endometriosis-related ovarian cancer, especially ovarian CCC, which exhibits unique characteristics.

## 5. Genomic and Epigenetic Alterations in Ovarian CCC

How is the expression of the ovarian CCC signature regulated? A comprehensive DNA methylation analysis showed that CCC had a profile distinct from that of other histological subtypes [14,43]. This profile involved many genes involved in *HNF1* transcription. Both expression and DNA methylation analyses revealed that the *HNF1B* signal is activated through decreased methylation in ovarian CCC. In contrast, ERa signaling is activated in endometrioid carcinoma of the ovary. In our previous report, whole exome sequencing using ovarian CCC revealed genomic abnormalities in many signals, as well as genes belonging to the *HNF1B* signal in ovarian CCC [44]. Mutations and amplification of *PIK3CA* were found to occur in 54% and 26% of cases, respectively, and mutations and amplification of *KRAS* were identified in 10% and 18% of ovarian CCC cases, respectively [44]. Genomic abnormalities in *KRAS-PI3K* signaling were identified in 82% of ovarian CCC cases; furthermore, 79% of CCC cases show copy number alterations in *MYC-RB* signaling, including *MYC* amplification and *RB1* deletion in 81% and 31% of CCC cases, respectively [44]. The *TP53* gene was deleted, and *MDM2* and *MDM4* were amplified in ovarian CCC. *ARID1A* mutations were identified in 62% of cases, and *SMARCA2* and *SMARCA4* were deleted in 21% and 41% of CCC cases, respectively [44]; moreover, genes belonging to the SWI/SNF complex exhibited genomic abnormalities in 85% of the CCC cases. These findings suggest that the activation of *KRAS-PI3K* signaling, *MYC-RB* signaling, and the suppression of the *TP53* network and the SWI/SNF complex are associated with ovarian CCC [44]. Other studies have also identified similar frequent genomic alterations in ovarian CCC. In total, 40–62% and 33–51% of CCC cases show *ARID1A* and *PIK3CA* mutations [45,46]. *KRAS* mutations are identified in 9–17% of cases, and *TP53* mutations are observed in 5–15% of ovarian CCC cases [45,46,47,48]. The frequency of single-nucleotide variants was significantly lower in ovarian CCC than in high-grade serous carcinoma, whereas the ratio of whole-arm copy number variants (CNVs) among all CNVs in CCC was significantly higher than that in high-grade serous carcinoma [49]. Chromosome 20q13.2, including the *ZNF217* (Zinc finger protein 217) locus, is frequently amplified in ~36% of CCC cases [50]. Amplifications of the *MET* (chr7q31) and *AKT2* (chr19q13.2) oncogenes have been identified in 31% and 24% of ovarian CCC, respectively, and deletions of the *CDKN2A/2B* (cyclin-dependent kinase inhibitor 2A/2B) tumor suppressor genes have been detected in 17% of cases [50,51,52]. We confirmed the protein levels of activated signals in ovarian CCC. HNF1B is highly expressed, and ERa is poorly expressed in ovarian CCC [30,31]. The expression of pAKT, which reflects *PI3K* signaling, is high, and pMAPK, which reflects *KRAS* signaling, is also highly expressed, similar to the genomic characteristics. Based on the immunohistochemistry of nine subunits belonging to the SWI/SNF complex, 61% of the cases exhibited the loss of at least one SWI/SNF complex subunit; thus, genomic and epigenetic abnormalities occur in ovarian CCC, which differs from the other histological types.

## 6. Regulation of the Unique Characteristics and Mitochondrial Metabolism by HNF1B in Ovarian CCC

From the distinct gene expression profile of ovarian CCC compared to other histologic subtypes [32,49], HNF1B is the most abundantly upregulated transcription factor in CCC [32] and in 40% of endometriotic cysts without malignancy [53]. The OCCC signature contained many genes related to oxidative stress, glucose metabolism, and *HNF1B* transcript signaling, as described above [32]. *HNF1B* mutations cause maturity-onset diabetes in the young type five [54]; therefore, we next considered the influence of HNF1B on metabolism in CCC.

HNF1B induces glucose uptake into ovarian CCC cell lines through glucose transporter 1 (GLUT1) and activates glycolysis flux and lactate secretion [55]. Comprehensive metabolome analysis using the HNF1B-suppressed CCC cell line revealed that HNF1B increased lactic acid levels and decreased citric acid levels, indicating that anaerobic metabolism is enhanced and the TCA cycle is decreased [56]. This phenomenon commonly occurs in cancer cells and is called the Warburg effect. The Warburg effect, first reported in 1956, refers to the enhanced anaerobic metabolism in cancer cells, even under aerobic conditions [57]; however, the reason for the Warburg effect in cancer cells remains controversial.

HNF1B also contributes to glutathione synthesis [50]. An HNF1B-suppressed CCC cell line showed increased ROS and higher sensitivity to cisplatin compared to an HNF1B-retained CCC cell line [56]. This regulation was induced by glutathione synthesis through the cystine transporter SLC3A1, regulated by HNF1B [56]. HNF1B also regulates glutathione levels in renal cell carcinoma [58]. *ARID1A* deficiency, which is identified in approximately half of ovarian CCC cases, suppresses glutathione synthesis. *ARID1A*-deficient cancer cells are specifically vulnerable to inhibitors of the antioxidant glutathione metabolic pathway and the glutamate–cysteine ligase synthetase catalytic subunit, a rate-limiting enzyme for glutathione synthesis [59]. These results imply that antioxidant glutathione synthesis is regulated by HNF1B and *ARID1A* deficiency.

HNF1B suppresses cell proliferation [60] and represses cyclin D1 through direct suppression of SMAD6 expression, which is repressed by CDKN2A as an HNF1B-interacting protein [60]. HNF1B induces aberrant retention of the G2 checkpoint, indicating chemoresistance of ovarian CCC [61]. These functions are compatible with the clinical behavior of the slow growth of ovarian CCC.

Why does the Warburg effect occur in ovarian CCC? Aerobic metabolism occurs in the mitochondria (Figure 1), which produce energy in an efficient electron transfer system by regulating the TCA cycle. However, when mitochondria work, they generate oxidative stress through the Fenton reaction using catalysts called iron–sulfur clusters [62]. Cells maintain a balance between energy efficiency and oxidative stress generation regulated by mitochondria in order to survive (Figure 1). Mitochondria are, so to speak, a “double-edged sword”. In CCC cells, anaerobic metabolism is activated, whereas aerobic metabolism is suppressed, suggesting the suppression of mitochondrial function (data in press). It is thought that they deliberately use anaerobic metabolism, which is less energy efficient, to survive in stressful environments. Mitochondrial metabolism enhances chemosensitivity in human ovarian cancers [63]. The reason why they use inefficient anaerobic metabolism is that the suppression of mitochondrial function suppresses oxidative stress in mitochondria, which is thought to be related to chemoresistance. Mitochondria play a crucial role in ferroptosis [64,65,66]; therefore, we believe that resistance to chemotherapies and ferroptosis could be overcome by activating mitochondrial function as a novel attractive therapeutic target in ovarian CCC. Targeting mitochondria itself using iron chelator results in the inhibition of the proliferation and induction of cell death via interfering with the iron metabolism and destabilizing the mitochondria in cancer cells [67].

In summary, ovarian CCC develops in endometriotic cysts, which involves an environment of iron-induced persistent oxidative stress. In this stressful environment, genomic and epigenetic abnormalities occur, and ovarian CCC signatures, including HNF1B, are overexpressed. HNF1B contributes to the stress resistance and Warburg effect which are characteristic biology of ovarian CCC. These findings indicate that adaptation to the microenvironment of endometriotic cysts leads to high expression of ovarian CCC signatures, especially *HNF1B*, resulting in CCC (Figure 2).

## 7. Hypothesis—Evolution or Reversion of Mitochondrial Metabolism in Ovarian CCC

Life has evolved for energy homeostasis. The earth was formed ~4.6 billion years ago, and is defined by four eons, including the Hadean, Archean, Proterozoic, and Phanerozoic eons [68]. The Hadean eon, named after Hades, was inhospitable because of the excessively hot and anoxic conditions. Life began 3.8 billion years ago in the Archean eon [68]. In the Archean eon, which was filled with warm and ferruginous (anoxic and Fe^2+^ dominated) oceans, other than possibly a few “whiffs”, atmospheric O_2_ was essentially nil until the great oxidation event (GOE) 2.3 billion years ago (Figure 3). The Proterozoic eon began 2.5 billion years ago; atmospheric oxygen increased several times to ~2%, while the oceans remained essentially anoxic in the GOE. In the Proterozoic eon, eukaryotes first appeared 1.5 billion years ago in euxinic oceans and developed until O_2_ production by oxygenic cyanobacteria and plants ~600 million years ago. Although oxygen production increased tremendously by the combined activity of cyanobacteria and plants, the oxygen was immediately reduced by the large amounts of H_2_S and Fe^2+^; subsequently, the oceans were oxidized, atmospheric oxygen increased to the present values, and led to an end to H_2_S as an energy source. In the Phanerozoic eon, the earth is supported by solar energy and an abundance of atmospheric oxygen. Mitochondria have evolved to give organisms the energy to exploit new, more complex morphologies interacting in increasingly complex ecosystems. Eukaryotic organisms with nuclei acquire mitochondria to use oxygen to efficiently produce energy; however, the abundance of oxygen poses a new threat to life. Mitochondria simultaneously emit oxidative stress which animals are forced to deal with. H_2_S-dependent animals either became extinct, retreated to isolated sulfide niches, or adapted. Mass extinctions were often associated with a fall in ambient O_2_ and an increase in H_2_S, perhaps providing a biological filter for descendants that retained some degree of tolerance to hypoxia and sulfide.

Does ovarian CCC use mitochondria for evolution or reversion? Charles Darwin hypothesized that all species of organisms arise and develop through the natural selection of inherited variations that increase an individual’s ability to compete, survive, and reproduce—called Darwinian theory [69]. Lamarck’s theory contrasts with Darwinian theory of evolution by natural selection. Lamarckism is based on the belief that the environment produces changes in biological structures, and that the degree of use or disuse of biological structures shaped evolution along with the inheritance of acquired adaptability [70]. When ovarian CCC is viewed as a mitochondrial metabolic disease, the evolutionary theory of Lamarck can better explain cancer biology than the Darwinian theory. Because ovarian CCC needs to develop in stressful environments, they do not want to generate more ROS. Therefore, CCC cells suppress mitochondrial functions in order to survive in harsh environments, probably leading to resistance to oxidative stress and ferroptosis. This hypothesis might indicate the evolution or reversion of mitochondrial metabolism in ovarian CCC.

## 8. Conclusions

Our study, along with others, suggests two carcinogenic processes in endometriotic cysts. The first is carcinogenesis under persistent stressful environments, resulting in the carcinogenesis of ovarian CCC, because endometriotic cysts contain abundant oxidative stress through free iron. The second potential carcinogenesis process is primarily estrogen-dependent and gives rise to endometrioid carcinoma. In ovarian CCC cells, HNF1B plays crucial roles in establishing the unique characteristics of ovarian CCC, including morphology, slow growth, and resistance to conventional chemotherapies. Abundant glycogen accumulation in the cytoplasm is caused by the uptake of glucose, which increases anaerobic glycolysis activity by HNF1B. HNF1B suppresses the cell cycle, which is compatible with the clinical growth of ovarian CCC. HNF1B contributes to resistance against anti-cancer agents through glutathione synthesis and the Warburg effect. In the Warburg effect, mitochondrial function is suppressed to decrease ROS production. The purpose of life evolution is the efficient production of energy. Although eukaryotes require mitochondria to generate energy efficiently, ovarian CCC cells suppress mitochondrial function. A better understanding of the fundamental biology of ovarian CCC and innovative strategies targeting metabolism that overcomes evolution are necessary to develop useful therapies against ovarian CCC.

## Figures and Tables

**Figure 1 cancers-13-02413-f001:**
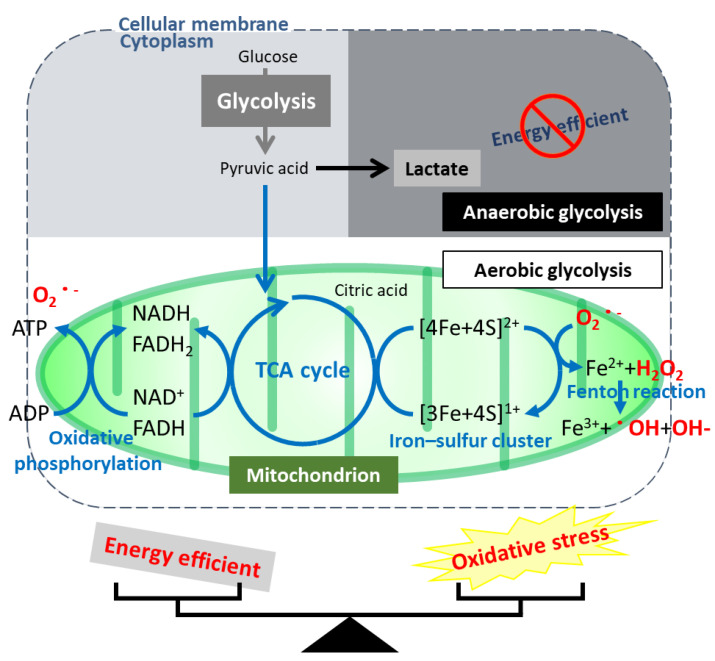
Mitochondrion is a “double-edged sword”. The balance between the state of energy efficient and oxidative stress in a mitochondrion.

**Figure 2 cancers-13-02413-f002:**
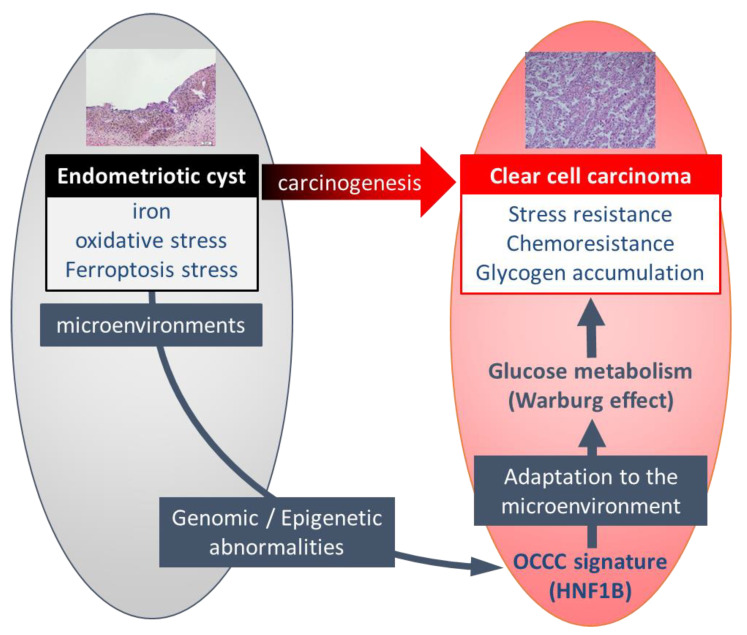
The schema of influence of the microenvironment of endometriotic cyst on the characteristics of ovarian clear cell carcinoma.

**Figure 3 cancers-13-02413-f003:**
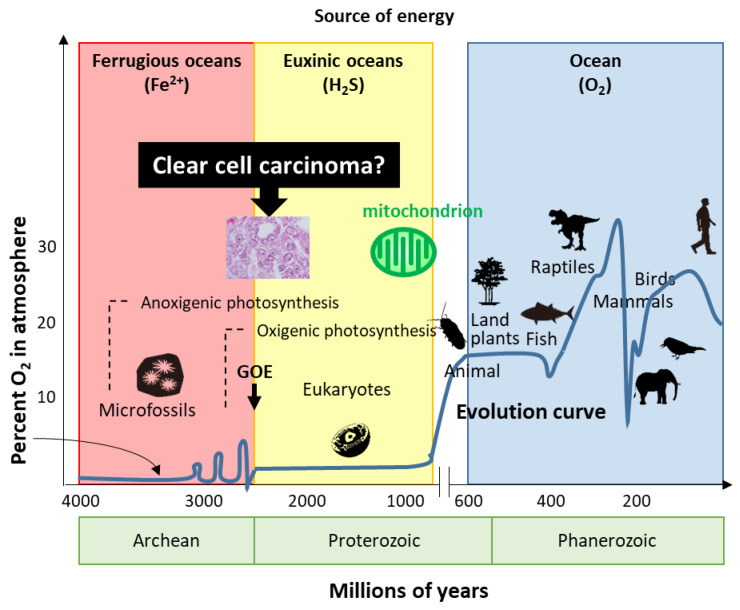
Evolution of life. Based on energy production, ovarian clear cell carcinoma seems to show evolution or reversion because of traditional energy metabolism. GOE: great oxidation event.

## Data Availability

Data available in a publicly accessible repository.

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
