# Peer review of "Acquired Evolution of Mitochondrial Metabolism Regulated by HNF1B in Ovarian Clear Cell Carcinoma"

_cancers, 2021, doi:10.3390/cancers13102413_

Round 1

Reviewer 1 Report

The review is well-written and thorough.  I would only ask that the authors define rs11651735 for those readers who would learn from the review but don't already know what this is.

Author Response

Reviewer 1

The review is well-written and thorough. I would only ask that the authors define rs11651735 for those readers who would learn from the review but don't already know what this is.

>>> We appreciate your suggestion. We edited to ” single nucleotide polymorphisms (rs11651735)” in the Introduction and the section 2.

Reviewer 2 Report

None

Author Response

None

>>> We appreciate all acceptance of our article.

Reviewer 3 Report

The review manuscript well described the unique molecular detail of ovarian clear cell subtype, which is highly resistant to chemotherapy. This CCC subtype is seriously under studied compared to a high grade serous subtype. Any unique signatures at molecular and cellular levels are highly significant to uncover drug candidate. 

The section 1 to 6, authors have provided interesting scientific sound to support questions that are specifically observed from many studies. However, section 7 is somehow weak statement which is likely more speculative rather than being focusing on strong scientific rationale.

I think the author should emphasis more about the role of iron and possible using iron chelating agent to compromise the mitochondrial function.

Author Response

The review manuscript well described the unique molecular detail of ovarian clear cell subtype, which is highly resistant to chemotherapy. This CCC subtype is seriously under studied compared to a high grade serous subtype. Any unique signatures at molecular and cellular levels are highly significant to uncover drug candidate.

The section 1 to 6, authors have provided interesting scientific sound to support questions that are specifically observed from many studies. However, section 7 is somehow weak statement which is likely more speculative rather than being focusing on strong scientific rationale.

I think the author should emphasis more about the role of iron and possible using iron chelating agent to compromise the mitochondrial function.

>>> Thank you for your informative suggestion. As the reviewer points out, section 7 is a weak statement because section 7 is composed of our hypothetical descriptions. The standpoint of this section is distinct from the previous six sections. We added “Hypothesis” in the title of this section in order to clarify what we want to state. We also added descriptions that the possible targeting iron on mitochondrial function in the section 6.

Reviewer 4 Report

Interest subject about ovarian cancer and its development from endometriosis. But I would suggest some improvements: The review mentioned the gene PIK3CA has mutations in CCC, but it does not mention in detail what roles plays in CCC as the other described genes. It would be helpful to add more information if available. The second improvement, the introduction and the first four parts of the review present three questions about CCC and they are not fully answered in those sections. The complete responses are in section 6 and 7. I would suggest reorganizing the information and add the section 6 and 7 to the section 2, 3 and 4.

Author Response

Interest subject about ovarian cancer and its development from endometriosis. But I would suggest some improvements: The review mentioned the gene PIK3CA has mutations in CCC, but it does not mention in detail what roles plays in CCC as the other described genes. It would be helpful to add more information if available. The second improvement, the introduction and the first four parts of the review present three questions about CCC and they are not fully answered in those sections. The complete responses are in section 6 and 7. I would suggest reorganizing the information and add the section 6 and 7 to the section 2, 3 and 4.

>>> We appreciate your informative suggestions. We added description about PIK3CA mutations as well as KRAS mutations in the section 2. As Reviewer 3 also pointed out, section 7 looks somehow weak statement. We really agree with your complaints because section 7 is composed of our hypothetical descriptions. The standpoint of this section is distinct from the previous six sections. We added “Hypothesis” in the title of this section in order to clarify what we want to state.

Round 2

Reviewer 3 Report

There are small adding/deleting to make the manuscript well read.

Author Response

We have edited the manuscript based on the suggestions and explained why we did not edit some questions. We hope that our responses satisfy the editor and reviewers.

Reviewers’ comments

Comment 1

Ovarian CCC?

>>> We appreciate your suggestion. Actually endometriotic cysts is the risk of ovarian cancer, especially CCC and endometrioid carcinoma. We edited it.

Comment 2

In Japan

>>> Ovarian cancer is the worst gynecologic malignancy in Japan as well as the USA and western countries. Therefore we did not add “in Japan”.

Comment 3

Delete this. Highlighted “in Japan”

>>> CCC is the second most common ovarian cancer in Japan, but the third most common all over the world. Therefore, we did not delete “in Japan”.

Comment 4

Are there four questions? Not three?

>>> They look like four questions. But ”why is endometriosis-related ovarian cancer more associated with specific histological subtypes, including CCC and endometrioid carcinoma?” and “why distinct histological subtypes arise in endometriotic cysts?” are described in the same section. Therefore we proposed three clinical questions according to the sections in order to allow readers easily recognition.

Comment 5

Ovarian CCC

>>> We edited it.

Comment 6

Where are the references?

>>> We apologize for our fault. We added references.
